# Immediate Tooth Autotransplantation with Root Canal Filling and Partially Demineralized Dentin/Cementum Matrix into Congenital Missing Tooth Region

**DOI:** 10.3390/jfb13020082

**Published:** 2022-06-16

**Authors:** Masaru Murata, Yasuhiro Nakanishi, Kaoru Kusano, Yukito Hirose, Junichi Tazaki, Toshiyuki Akazawa, Itaru Mizoguchi

**Affiliations:** 1Division of Regenerative Medicine, School of Dentistry, Health Sciences University of Hokkaido, Hokkaido 061-0293, Japan; junichi1125@icloud.com; 2Division of Fixed Prosthodontics and Oral Implantology, School of Dentistry, Health Sciences University of Hokkaido, Hokkaido 061-0293, Japan; nakanisi@hoku-iryo-u.ac.jp (Y.N.); yukito@hoku-iryo-u.ac.jp (Y.H.); 3Department of Oral Implantology, Osaka Dental University, Osaka 573-1121, Japan; kusano-k@cc.osaka-dent.ac.jp; 4Industrial Technology and Environment Research Development, Hokkaido Research Organization, Hokkaido 060-0819, Japan; akazawa-toshiyuki@hro.or.jp; 5Division of Orthodontics and Dentofacial Orthopedics, Graduate School of Dentistry, Tohoku University, Sendai 980-8575, Japan; itaru.mizoguchi.c3@tohoku.ac.jp

**Keywords:** autograft, bone, dentin, DDM, human, immediate, tooth, transplantation

## Abstract

This clinical report describes immediate tooth auto-transplantation with an autograft of partially demineralized dentin/cementum matrix (pDDM), based on an orthodontic treatment plan for a 16-year-old male patient with a congenital missing tooth (#45). First, vital teeth (#14, #24) were extracted, and root canal filling (#14) was immediately performed with the support of a fixation device. Simultaneously, the tooth (#24) was crushed in an electric mill for 1 min, and the crushed granules were partially demineralized in 2% HNO_3_ solution for 20 min as the graft material. Next, the donor tooth was transplanted into the created socket (#45), and stabilized using an enamel bonding agent. The wet pDDM was loaded into the location of the congenital missing tooth, and the flap was repositioned. The bonding agent for stabilization was removed at 28 days, and also small contact points between the transplanted tooth and the upper premolar (#14) were added using photopolymerizable composite resin. X-ray photos were taken sequentially, and there were no postoperative complications. The radiographic images showed that the periodontal ligament space and alveolar ridge line could be observed at 18 months. The pDDM was harmonized with the mandible, and the remodeled bone-like shadow was observed in the graft region. We concluded that immediate tooth transplantation with root canal fillings and autogenous pDDM may be a valuable alternative to dental implanting or bridge formation for patients with a congenital missing tooth, followed by orthodontic treatment.

## 1. Introduction

Tooth autotransplantation is a well-established surgical procedure for the replacement of missing or unrestorable teeth [1,2]. The survival and success rates of transplanted teeth varies, depending on the type of transplanted tooth, stage of root development of donor teeth, surgical technique, the follow-up period, and methodology of the research [2,3]. It is well known that the viability of the periodontal ligament cells after extraction of the donor tooth is most important aspect for the success of tooth transplantation [4,5,6]. The transplanted tooth with complete root formation is rarely expected to revascularize and reinnervate [7]. For complete root teeth, the success rate of tooth autotransplantation after root canal filling (RCF) was higher than that of teeth without RCF [8]. Thus, endodontic treatments will be required before or after the surgery, to prevent apical inflammation derived from pulp necrosis [8,9,10]. Autotransplantation of premolars has become a reliable treatment option in cases with certain orthodontic problems [11]. Interestingly, the transplantation of a mature third molar for the immediate closure of the oroantral communication after molar extraction was successfully reported as a challenging procedure [7].

This clinical report describes autotransplantation of a premolar with complete roots, simultaneously with immediate RCF and autograft of a partially demineralized dentin matrix (pDDM), based on the orthodontic treatment for a 16-year-old male patient with a congenital missing tooth. Both the tooth-fixing device for protection of the periodontal ligament cells and the mill for the pDDM preparation contribute to a new dental procedure for immediate tooth transplantation and towards the quality of life of patients. 

The aim of this study was to evaluate the radiographic evidence after autotransplantation of a tooth, with immediate RCF and partially demineralized dentin matrix (pDDM) granules, into the missing tooth region using the support of a newly developed tooth-fixing device and an electric mill. The radiographic evidence highlighted that the periodontal ligament space and alveolar ridge line could be observed at 18 months after the simultaneous and immediate tooth autotransplantation with RCF and pDDM.

## 2. Materials and Methods

### 2.1. Clinical Case

A 16 year-old-male was introduced from a private orthodontic dental clinic to our dental unit in 2011. The patient presented with a congenital missing lower left second premolar (#45), examined on initial panoramic X-ray photo (Figure 1). He had lost a milk tooth (#4E) a few weeks previously. Clinical diagnosis revealed a congenital missing tooth (#45), crowding, and maxillary protrusion. His medical history was unremarkable.

### 2.2. Surgical Planning

Based on examinations, including computed tomography (CT), the surgical treatments were planned in 2012, as follows: Distance between alveolar crest and mandibular canal: 19 mm. Upper second premolars (#14, #24): length 15mm, orthodontic extraction. The right premolar (#14): immediate RCF and transplantation into #45. The left premolar (#24): pDDM graft for bone augmentation.

### 2.3. Extraction and Immediate Root Canal Filling

First, vital-teeth (#14, #24) were extracted using forceps (Figure 2A) under local anesthesia. Tooth #14 was set in a tooth-fixing device (Fix^®^, WiSM Mutoh Co., Ltd., Tokyo, Japan) (Figure 2B), which could protect periodontal ligament cells during the endodontic treatments (Figure 2C), and the 2 root canals were filled using gutta percha points (Gutta percha soft points^®^, GC Co., Ltd., Tokyo, Japan) and sealer (Dentalis^®^, Neo Dental Chemical Products Co., Ltd., Tokyo, Japan) immediately, with a device using a rubber dam for transplantation (Figure 2D). The extraoral time of the endodontic treatments was about 10 min. 

### 2.4. Preparation of Partially Demineralized Dentin/Cementum Matrix (pDDM)

Simultaneously, tooth #24 was crushed with saline ice blocks in a Zirconium (ZrO_2_) vessel with ZrO_2_ blade at 12,000 rpm for 1 min using a mill (OSTEO-MILL^TM^, WiSM Mutoh Co., Ltd., Tokyo, Japan) (Figure 3A–C). The crushed tooth granules (Figure 3D) were immediately demineralized in 1.0 L of 2.0% HNO_3_ solution for 20 min. The pDDM granules including cementum were extensively rinsed in cold distilled water (D.W) (Figure 3E).

### 2.5. Tooth Transplantation and pDDM Graft

A full-thickness mucoperiosteal flap was elevated, and the drill of the dental implant system (Frialit-2^®^, Dentsply Sirona Co., Ltd., New York, NY, USA) was applied for the enlargement of the transplant cavity (#45). Cortical perforations were performed into the dense mandibular bone. The donor tooth (#14) after RCF and the removal of a part of crown portion was transplanted atraumatically into the cavity (Figure 4A), and was stabilized with the lower left molar (#46) using enamel bonding agent (Super-Bond^®^, SUN MEDICAL Co., Ltd., Tokyo, Japan). Then, pDDMs were loaded onto the places on the mandible around the transplanted tooth (Figure 4B). The transplanted tooth was not in contact with the upper teeth. After 5 min, blood coagulation including pDDM was confirmed (Figure 4C), and the flap was repositioned and sutured with nylon threads (Figure 4D). The patient took cephem antibiotics (Cefaclor 250 mg, 3 times/day; Kefral^®^, Osaka, Japan) for only 3 days. The nylon threads were removed at 10 days after the surgery. The bonding agent for stabilization was removed at 28 days, and also slight contact points between the transplanted tooth and the upper premolar (#14) were added using photopolymerizable composite resin (SOLARE P^®^, GC Co., Ltd., Tokyo, Japan).

### 2.6. Radiographic Evaluation

Initial X-ray photos including CT were taken for overall diagnosis. Next, panoramic and dental X-ray photos were taken just after the surgery (Figure 4), and then the root and periodontal ligament space were evaluated using clinical and X-ray photos (Figure 5, Figure 6, Figure 7, Figure 8 and Figure 9).

## 3. Results

### 3.1. Gross View and Examinations after Surgery

One day after the surgery, a blood clot was formed in the pocket around the transplanted tooth (Figure 5A,B). Gingival inflammation at the recipient site was slight, the orthodontic wire with brackets was set in the maxillary teeth, and the orthodontic treatment started at 7 days (Figure 5C). The gum healed normally at 10 days, and the threads were removed (Figure 5D). The grafted pDDM granules were not exposed and were received by the periodontal tissues. At 4 weeks, the tooth fixation was removed, and occlusal contact was created using composite resin. The orthodontic treatments continued for 2.5 years in the private dental clinic. The orthodontic movement and rotation were possible for the transplanted tooth, and a metallic sound was not heard at 1.5 years using a percussion test. At 2.5 years, the periodontal pocket was approximately 2 mm. During the entire follow-up period of 4.5 years, complications such as abnormal mobility did not occur.

### 3.2. Radiographic Evaluation

Just after the surgery, the extracted sockets of the maxillary premolars and the root canal filling of the transplanted tooth were seen clearly (Figure 6A). The grafted pDDM were confirmed as each radiopaque granule (Figure 7A). The transplanted root and bone showed a similar radio-opacity, and a periodontal ligament space was not seen (Figure 6A). After 1.5 year of orthodontic treatment, the periodontal ligament space and alveolar ridge line were observed, and pDDM granules were not seen clearly (Figure 6B and Figure 7B). In addition, all wisdom teeth were already extracted (Figure 6B). A slight curve of the root surface was observed adjacent to a normal periodontal ligament space and the lamina dura during the orthodontic treatments (Figure 7B). The view at 2.5 years shows removal of orthodontic devices, except the mandibular retention wire, and a single standing of the transplanted tooth without wire (Figure 6C and Figure 7C). A continuous periodontal ligament space and lamina dura were found, and root concrescence did not occur (Figure 7C and Figure 8B).

## 4. Discussion

### 4.1. Immediate System for Tooth Transplantation with Root Canal Filling (RCF) and pDDM

It is well known that a key of a successful tooth autotransplantation is the viability of periodontal ligament cells damaged by the extraction [4,5,6]. Therefore, in Japan, the extracted tooth without root canal treatments is transplanted into the socket as soon as possible, fixed with wire or bonding, and then endodontic treatments will start within 1 month, conventionally [7]. The transplanted tooth is subsequently checked with clinical and radiographic evaluations. We considered that a cornerstone of successful tooth autotransplantation was to develop a device that could protect periodontal ligament cells from bacteria, debris, and drying out during complicated root canal treatments. Dental doctors know the difficulties of endodontic treatments in the oral area, especially for the transplanted tooth, because the tooth itself remains unstable and endodontic treatments are very complicated with several curved and/or calcified canals. If RCF is not performed well for teeth with complete roots, unfortunately tooth autotransplantation will fail [8]. It is easier to perform RCF in the extraoral than in the intraoral space.

Our unique tooth-fixing device (Fix^®^, WiSM Mutoh Co., Ltd., Tokyo, Japan) can protect periodontal ligament cells from debris and drying out during endodontic treatments, as shown in Figure 2; and RCF for two canals of a premolar (#14) was finished in 10 min. By using the device, we believe that periodontal ligament cells can be protected from risk factors, and the extraoral treatment time can be kept to a minimum.

Clinically, the immediate DDM autograft system requires speed, compared to a delayed autograft. It takes much more time to completely prepare a DDM (cDDM) than a pDDM. Human dentin/cementum granules after 2% HNO_3_, demineralization for 20 min, were compatible with 65–70% of demineralized human DDM [12]. Fortunately, bone regeneration using human dentin depends on the degree of demineralization and granule size [13]. The results revealed that 70% of the demineralized human DDM granules (size: 1.0 mm) had a better performance for bone formation than 100% of the demineralized DDM (cDDM) and non-demineralized dentin in rat calvarial bone defects [13]. Our team, therefore, developed an immediate autograft system of a partially demineralized dentin/cementum matrix (pDDM) using a new electric mill. The pDDM could be quickly prepared in 30 min with the support of the electric mill, while an immediate graft of pDDM was not performed before the development of the mill. Our system, using both the device and the mill, could contribute to decreasing the number and period of the dental treatments related to tooth transplantation.

### 4.2. Timing of Root Canal Filling (RCF)

The preservation of periodontal ligament cells is considered to be critical for the success of a transplanted tooth [4,5,6]. However, the timing of root canal treatments for donor tooth with complete roots has not been sufficiently discussed, nor well-documented [7,9,14,15]. Generally, in Japan, the extracted donor tooth is wrapped with wet gauze with normal saline solution during surgical procedures, transplanted into a socket, and then root canal treatments are started about 3–4 weeks later [7]. In India, root canal treatments were performed extra-orally for a mandibular third molar, and the transplanted tooth resulted in a satisfactory outcome [9]. In Taiwan, a postoperative root canal treatment resulted in a significantly lower extraction rate than preoperative or extraoral one, by using a nationwide population-based database [14]. The use of a rubber dam was recommended during postoperative root canal treatments, to improve the outcomes [14]. On the other hand, in Korea, it seems that RCF prior to the donor tooth extraction was recommended, except for the impacted teeth or ones were the access was impossible [15]. Additionally, a unique technique was also considered and reported in India [6]. The donor tooth after intraoral RCF was extracted and put in a customized reservoir, to reduce the extraoral time during the surgery [6]. In the present case, the extracted donor tooth was fixed using a newly developed device, and immediate RCF was performed in the device extra-orally, just after tooth extraction. The device can protect periodontal ligament cells from drying, debris, and bacteria during shaving of the tooth and complicated root canal treatments. More scientific research will be needed the to clarify which timing and condition is better in endodontic procedures for successful autotransplantation of a tooth with complete roots. 

### 4.3. Surgical Technique for Transplantation

Regarding the technique of autotransplantation of mature teeth, the extraoral root-end resection of teeth with complete root formation showed promising outcomes for transplants, especially with a single root canal and uncomplicated root morphology [16]. For formation of a transplant cavity, a donor tooth replica was fabricated before surgery using 3D printer machine [17]. The use of rapid prototyping for autotransplantation enabled accurate positional planning and decreased the extra-alveolar time and endodontic treatment rate [17].

### 4.4. Root Resorption Subsequent to Transplantation

Three types of root resorption after tooth autotransplantation are classified, as follows: surface resorption, inflammatory resorption, and replacement resorption [11]. Orthodontic rotation for autotransplanted premolars induced a slight surface resorption and a significantly shorter tooth length (mean 1.2 mm) [11]. In this case, a slight root surface resorption of the transplanted tooth was observed during orthodontic movement adjacent to a normal periodontal ligament space and the lamina dura (Figure 7B). The rotation might temporarily induce a limited surface resorption.

### 4.5. Dentin/Cementum as a Biological Graft Material

Collagen derived from cow or pig has been a commercially available organic material for medical use since the 1970s in Japan [18,19]. Autogenous pDDM is a composite of growth factors, inorganic and organic, with the original cross-links and has a natural structure [10]. As advantages, DDM possess the ability to coagulate platelet-free heparinized, citrated, and oxalated blood plasmas, and clotting constituents become denatured in contact with the insoluble coagulant proteins [20]. The coagulation action of blood plasma should be advantageous for hemostasis after surgical operations. DDM contains native growth factors such as bone morphogenetic proteins (BMPs) [21], and adsorbs several proteins derived from body fluids and BMP-2 [22,23,24,25]. Additionally, adhesive sequences in dentin/cementum collagen support the adhesion of mesenchymal cells as an anchorage matrix. The pDDM granule consists of the outer demineralized matrix (acid-insoluble collagen) layer and the inner non-demineralized layer in the structure. In this case, the grafted pDDM was harmonized with the mandible (Figure 7E,F), and the remodeled bone-like radio-opaque shadow was observed in the graft region (Figure 7B,C and Figure 8B). A limitation of this case is the lack of histological findings. A pDDM graft with a simultaneous tooth transplant was performed in this case. Therefore, an additional biopsy was never considered, due to ethical concerns for the patient. Several papers with histological evidence have been published showing that pDDMs were gradually absorbed and replaced by bone [23,26,27]. General limitations of tooth autotransplanting include the availability of a suitable donor tooth, sufficient volume of recipient bone, and matching them. 

### 4.6. Augmentation by Dentin/Cementum Materials 

Horizontal or vertical augmentation is a challenging field in bone regeneration. In this case, granules of pDDM were used for bone augmentation. In addition to granules, block, ring, and plate forms of pDDM can be prepared for bone regeneration [23,27,28]. In a rat onlay model, human DDM blocks with different 0.6N HCl-demineralization time (0, 10, 30, 60, 90 min) were evaluated histologically, and pDDM block demineralized for 30 min (pDDM/30) showed significantly increased new bone area than pDDM/10 at 8 weeks [29]. In adult rabbit onlay model, human DDM blocks of four different types (demineralized or non, with or without perforations) were evaluated histologically, and the DDM block with perforations revealed a better performance for bone augmentation than the other blocks [30]. In 2021, vertical augmentation was reported on the scratched skull without periosteum of adult-aged rats using human pDDM block [31]. As a pioneering case, horizontal augmentation using an autogenous pDDM block was successfully achieved on the anterior atrophic maxilla of a 56-year-old woman [32]. 

Based on dentin science and doctor’s ideas, dentin/cementum materials should advance bone regenerative therapy as highly acid-insoluble scaffolds for patients, in the near future.

## 5. Conclusions

The clinical and radiographic examinations showed that the autotransplant of a premolar (#14) with immediate RCF and pDDM graft into the congenital missing tooth region (#45) was successfully performed, to reconstitute the occlusion. The radiographic evidence highlighted that the periodontal ligament space and alveolar ridge line could be observed at 18 months after the immediate autotransplantation with RCF and pDDM. The tooth-fixing device and the electric mill contributed to enabling the immediate procedure for autotransplantation. A slight root surface resorption of the transplanted tooth might cause an orthodontic rotation.

We conclude that immediate tooth transplantation with RCF and autogenous pDDM should be a valuable alternative to a dental implant or bridge formation for patients with a congenital missing tooth followed by orthodontic treatment.

## Figures and Tables

**Figure 1 jfb-13-00082-f001:**
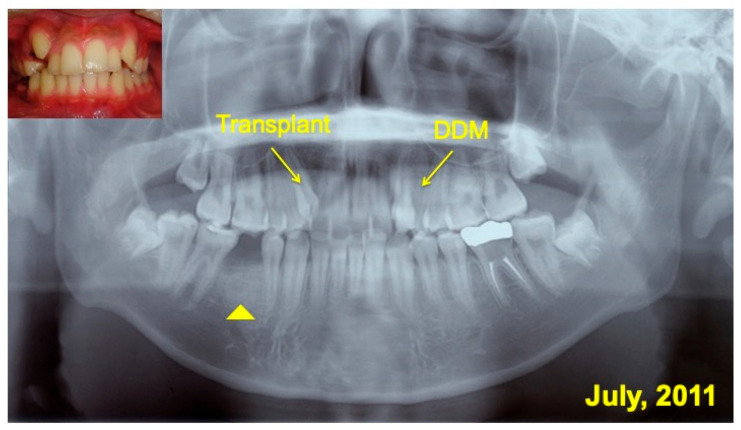
Initial views of panoramic X-ray photo and oral before orthodontic treatment. Arrows indicating #14 for transplant and #24 for pDDM. Arrowhead showing congenital missing tooth (#45).

**Figure 2 jfb-13-00082-f002:**
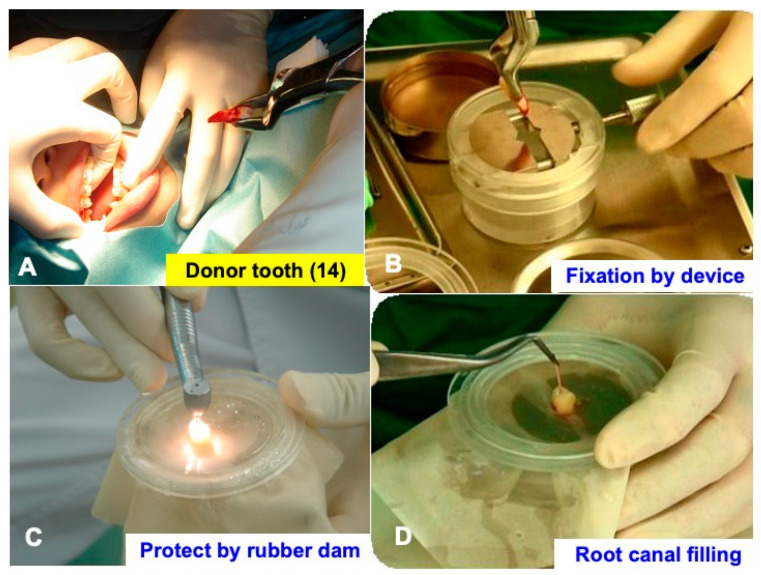
Immediate root canal filling in tooth-fixing device. (**A**) Extraction by forceps. Note: root covered with blood. (**B**) Setting with device. (**C**) Roots and periodontal ligament cells protected by rubber dam during treatments. (**D**) Immediate root canal filling (RCF).

**Figure 3 jfb-13-00082-f003:**
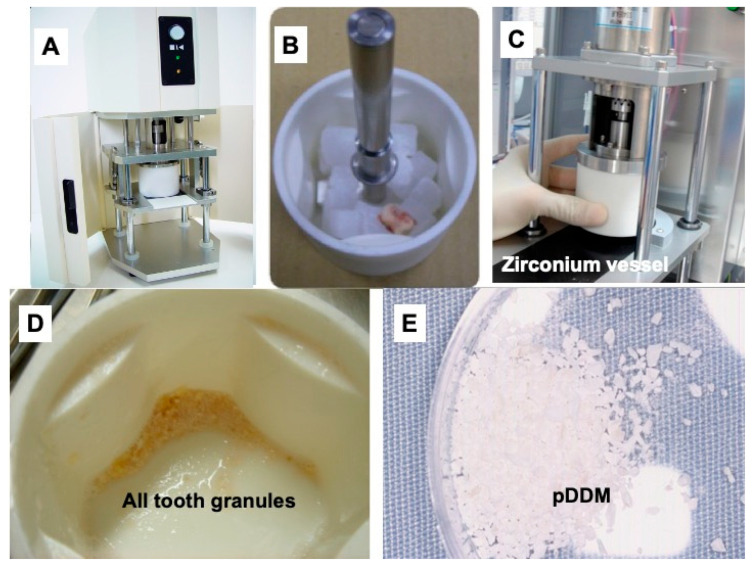
Preparation of partially demineralized dentin/cementum matrix (pDDM) using a mill. (**A**) View of entire mill. (**B**) Tooth on 0.9% saline ice blocks (1 × 1 × 1 cm^3^) in ZrO_2_ vessel. (**C**) Setting on stage. (**D**) All tooth granules crashed using ZrO_2_ blade in a ZrO_2_ vessel. (**E**) pDDM in dish. ZrO_2_: Zirconium.

**Figure 4 jfb-13-00082-f004:**
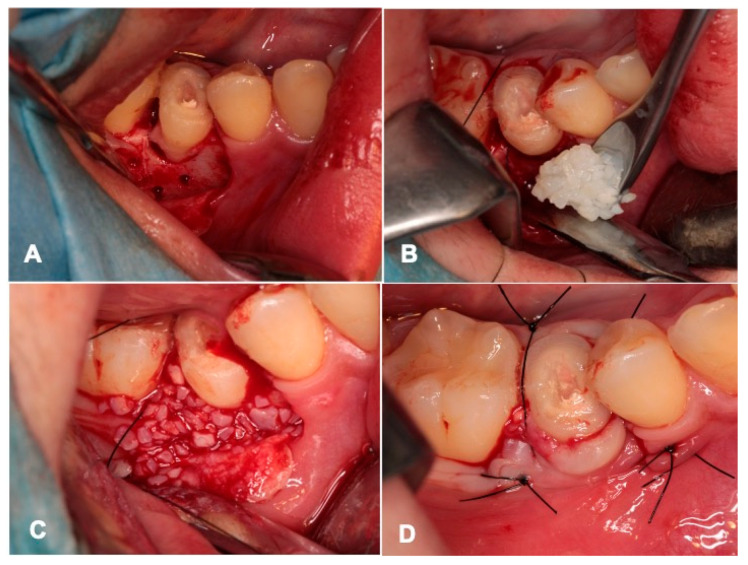
Surgical procedure. (**A**) Donor tooth (#14) after immediate RCF transplanted into cavity (#45). Note: unstable tooth and cortical perforations. (**B**) pDDM graft after tooth stabilization by bonding agent. (**C**) View of blood coagulation with pDDM. (**D**) Flap repositioned and sutured with nylon threads. Note: removal of a part of the crown.

**Figure 5 jfb-13-00082-f005:**
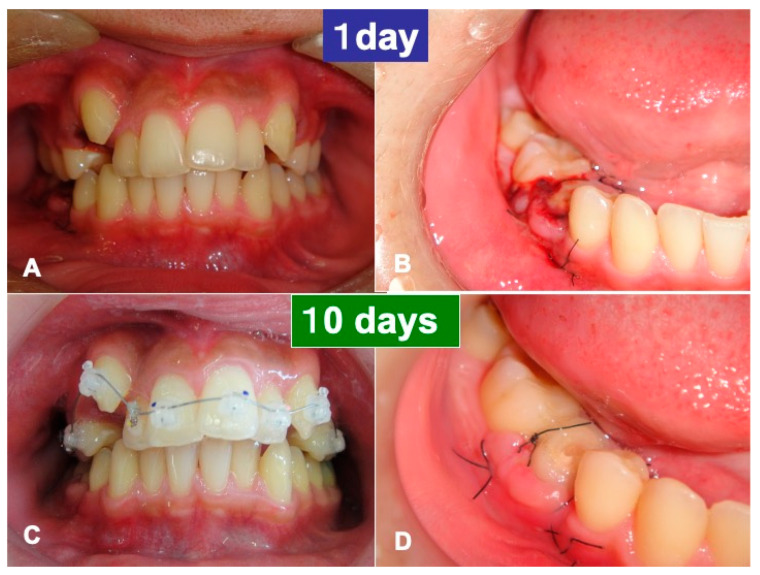
Gross views. (**A**) Whole view showing crowding. (**B**) Blood clot formation around the transplanted tooth at 1 day. (**C**) Orthodontic devices in the maxillary teeth at 10 days. (**D**) Normal healing of gum at 10 days. Note: pDDM covered with gum.

**Figure 6 jfb-13-00082-f006:**
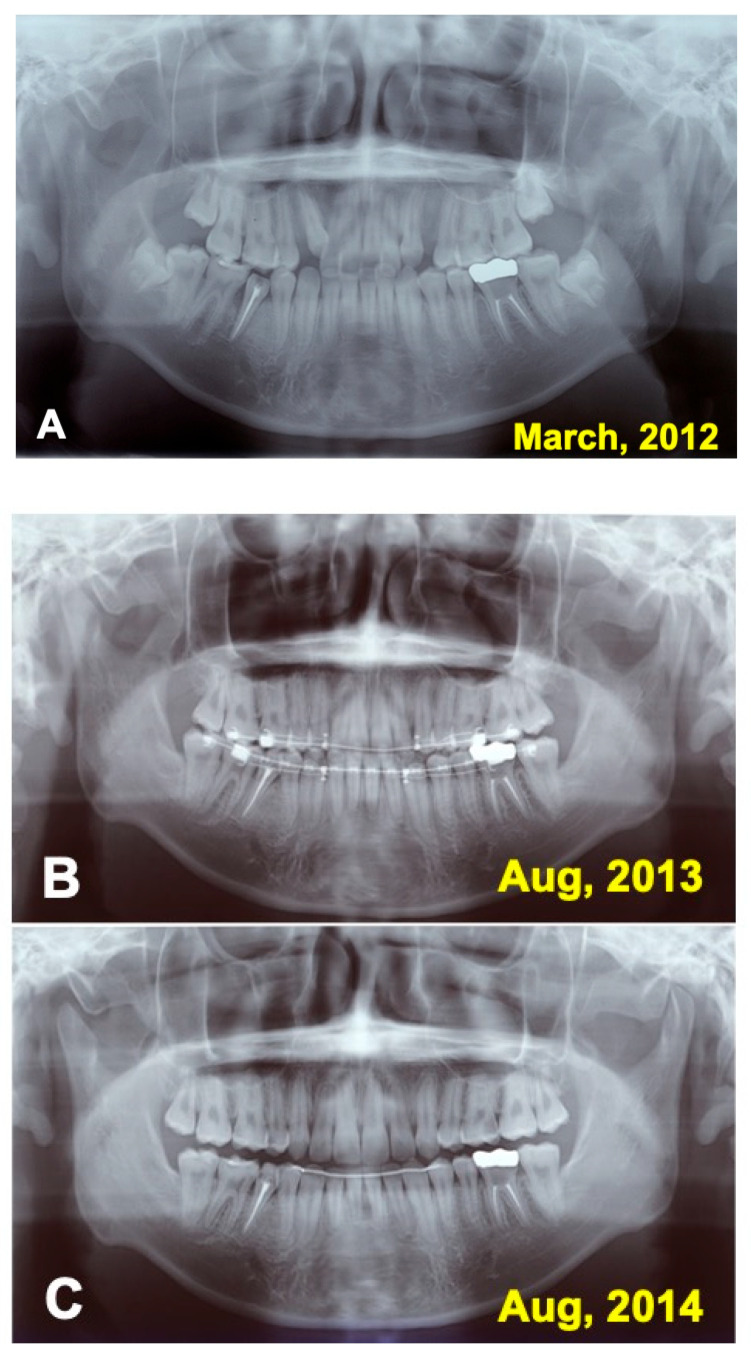
Panoramic X-ray photos. (**A**) Appearance just after surgery. (**B**) Appearance at 1.5 years during orthodontic treatment. Note: extraction of wisdom teeth (#38, #48). (**C**) Appearance at 2.5 years. Removal of orthodontic devices, except mandibular retention wire.

**Figure 7 jfb-13-00082-f007:**
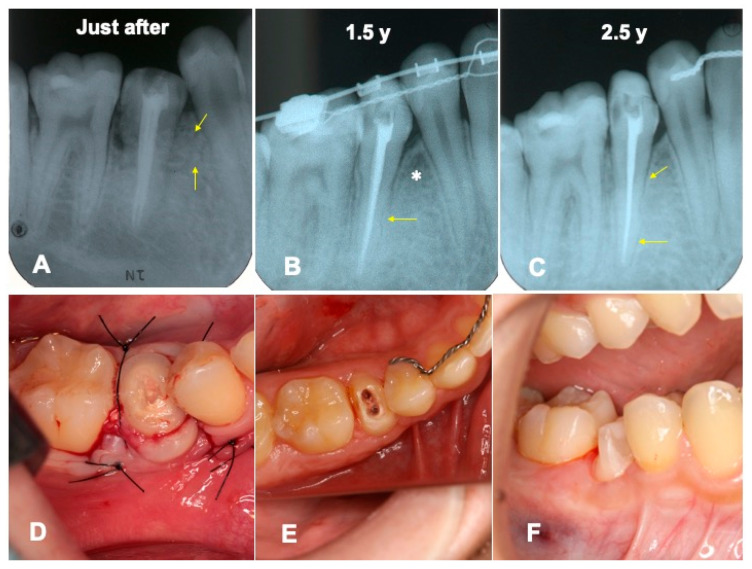
Dental X-ray and intraoral photos. (**A**) Appearance just after surgery. Arrows showing pDDM granules. (**B**) Appearance at 1.5 years. Arrow indicating suspected area after limited root resorption. Totally remodeled bone-like shadow (*). (**C**) Appearance at 2.5 years. Arrows indicating periodontal ligament space. (**D**) View just after immediate surgery. (**E**) View of root and periodontal tissues before being built up. (**F**) View just after core formation for impression of final ceramic crown. Abutment tooth built up with a direct resin composite core (BeautiCoreSystem^®^, Shofu Co., Ltd., Kyoto, Japan) with fiberpost (BeautiCoreFiberPost^®^, Shofu Co., Ltd., Kyoto, Japan).

**Figure 8 jfb-13-00082-f008:**
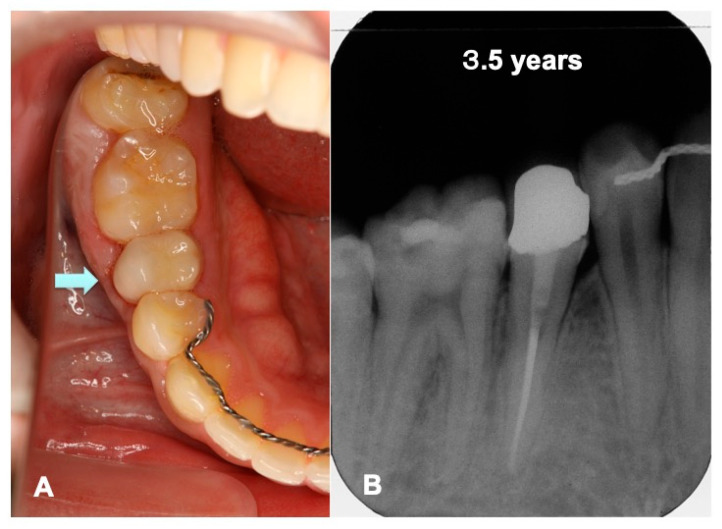
Set of partially veneered zirconia all ceramic crown (Nobel Procera^®^, Nobel Biocare Co., Ltd., Zurich, Sweden) rutted using adhesive resin cement (RelyX Ultimate^®^, 3M ESPE Co., Ltd., Seefeld, Germany) in 2015. (**A**) Mirror view just after final crown set. Arrow indicating biological gum line (#45). (**B**) X-ray photo showing continuous periodontal ligament space indicating non-ankylosis. Healthy bone-like shadow with lamina dura (#44–46) at recall follow-up after 3.5 years.

**Figure 9 jfb-13-00082-f009:**
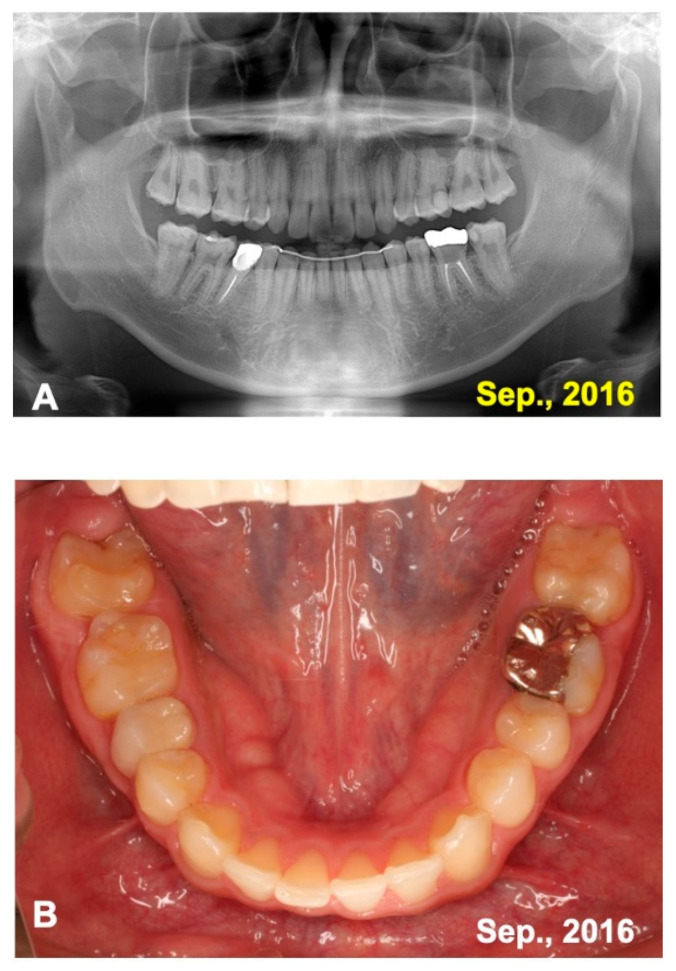
Final follow-up views in 2016. (**A**) X-ray photo showing continuous periodontal ligament space with lamina dura indicating non-ankylosis (#45). (**B**) View after removal of retention wire. Note: healthy gum line (#45).

## Data Availability

Data sharing is not applicable to this article.

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
