# Peer review of "Immediate Tooth Autotransplantation with Root Canal Filling and Partially Demineralized Dentin/Cementum Matrix into Congenital Missing Tooth Region"

_jfb, 2022, doi:10.3390/jfb13020082_

Round 1
Reviewer 1 Report
The manuscript by the Authors reports “Immediate Tooth Autotransplantation with Root Canal Filling and Partially Demineralized Dentin/Cementum Matrix into Congenital Missing Tooth Region”. This clinical case is under the scope of this Materials Journal, the topic is relevant for readers and this research deals with potentially significant knowledge to the field.
However, there are some concerns about the present manuscript:
(Keywords)
- Please order the keywords alphabetically for a standardised presentation of the keywords.
(Results)
- The clinical case has a follow-up, of 2.5 years, but the last control is from 2014…. It´s important to have a more recent control. And in this case, we have some root reabsorption after 5-years. Please, contact the patient for new control.
(Discussion)
- You need also discuss the hypothesis treatment plan of autotransplant of immature third molars for the first molar. The Dental papilla, when the root is formed, is in the apical zone, called the papilla apical (stem cells from apical papilla - SCAPs) that remain at the apical closure of the root. These stem cells are more immature than SHED or DPCS(in REPs animal study and a Clinical Case, read these references (doi.org/10.3390/APP9193942).
- When mentioning materials or devices: for some of them, you don't mention the manufacturer at all, for some you say only the manufacturer, for some the manufacturer and city, for some you mention the manufacturer and city/ country.
- Please, clarified other limitations of this study?
Author Response
We would like to resubmit our revised version, based on the guidance of your valuable advice.
This paper highlights tooth autotransplantation with immediate root canal filling (RCF) and autogenous partially demineralized dentin/cementum matrix (pDDM), based on orthodontic treatment plan for a 16 years old male patient with congenital missing tooth (#45). The novel surgical procedure was supported by tooth fix-device and tooth mill.

Reviewer 2 Report
Abstract
This clinical report describes autotransplantation of a premolar with complete roots, simultaneously with immediate root canal filling and autograft of partially demineralized dentin matrix.
Introduction
Line 39: “…surgical technique, the follow-up period, and methodology of the research [2,3].
The authors must refer to the article of the surgical technique in two phases:
Ferreira MM, Botelho MF, Carvalho L, Silva MR, Oliveiros B, Carrilho EV. Evaluation of dentin formed in autogenous tooth transplantation in the dog: a comparison between one- and two-stage surgical techniques. Dent Traumatol. 2012 Apr;28(2):97-100. doi: 10.1111/j.1600-9657.2011.01040.x. Epub 2011 Jul 14. PMID: 21752193.
Materials and Methods
Line 68: “Clinical diagnosis revealed congenital missing tooth (#45) and malocclusion.
What malocclusion?
Line 83, 84: The tooth#14 was set to a tooth-fixing device (Fix®, Tokyo Iken Co., Ltd) (Figure 2B) [9], which could protect periodontal ligament cells during the endodontic treatments…”
Why did the authors perform endodontic treatment after extraction, when this is not the procedure indicated in the bibliography? Did the tooth have 2 roots? what form? Were they divergent? What was the material used to fill the canals? How was the recipient socket prepared? What are the dimensions of the receptor alveolus?
Line 107: “…was stabilized with lower left molar (#46) by enamel bonding agent…”
The tooth has become immobile, which can lead to ankylosis. Why was this stabilization method used?
Line 111: “The patient took antibiotics only for 3 days.”
What is the antibiotic and how often are they taking it? Did you precribe other type of therapy?
Results
Fig. 7: It is mentioned that a nucleus and crown were made, but the reason and indication for having made a crown is not mentioned in the text.
Discussion
Why was “autograft of partially demineralized dentin matrix” used? What improved the success of the transplant?
Conclusions
Line 243-244:”We concluded that immediate tooth transplantation with RCF and autogenous pDDM should be a valuable alternative to dental implant or bridge formation for patients with congenital missing tooth followed by the orthodontic treatment”.
These conclusions are questionable and this article does not seem to me to have the scientific quality to be published.
Author Response
We would like to resubmit our revised version, based on the guidance of your valuable advice.
This paper highlights tooth autotransplantation with immediate root canal filling (RCF) and autogenous partially demineralized dentin/cementum matrix (pDDM), based on orthodontic treatment plan for a 16 years old male patient with congenital missing tooth (#45). The novel surgical procedure was supported by tooth fix-device and tooth mill.
1. Surgical technique was discussed in 4.2. Root Canal Filling (RCF) and Surgical Technique for Transplantation.
2. Drug, materials and all prosthodontics information were added in details.
Could you please check red sentences in our revised paper ?

Reviewer 3 Report
Dear authors,
the article covers a very interesting topic and I support its publication.
There is kind of a tendency to self-citation (7 articles by Murata!).
By the way I suggest some minor adjustements to improve the article for the readers.
Please specify if and which temporary filling has been used on the endodontic treatment. The absence of a coronal restoration may facilitate bacteria migration. The authors said that composite was placed in contact after 1 month. Was it placed also before but not in contact?
Line 106 the authors wrote: and was stabilized with lower 107 left molar (#46) by enamel bonding agent
Please specify brand, type and producer of the bonding agent.
Line 161-166: Was a post used to restore the endodontic-treated transplanted tooth?
If the coronal structure is available you could avoid to use a carbon-post (you could cite the following paper: Stress distribution in carbon-post applied with different composite core materials: a three-dimensional finite element analysis DOI 10.1080/01694243.2017.1304172)
Please specify in the text type of ceramic crown and restorative procedures. Captions in Figure 7 and Figure 8 shall be cited in text.
If lithium disilicate or zirconia or pfm (porcelain fused to metal) has been used please refer to an article about the importance of precision and seal of the restorative procedure: DOI 10.3390/polym13173002
In the discussion add a paragraph about orthodontic treatments as a first option in distributing and aligning teeth before taking into consideration implant dentistry or transplantations as well as a mean to obtain bone for implant purposes. Authors could highlight the sequence of preferred treatments: orthodontics, transplantation, orthodontics for bone regeneration for implants ( PMID: 19350058.), implants.
Author Response
We would like to resubmit our revised version, based on the guidance of your valuable advice.
This paper highlights tooth autotransplantation with immediate root canal filling (RCF) and autogenous partially demineralized dentin/cementum matrix (pDDM), based on orthodontic treatment plan for a 16 years old male patient with congenital missing tooth (#45). The novel surgical procedure was supported by tooth fix-device and tooth mill.
- Dental materials and prosthodontics procedures were described in details.
- DDM autograft was achieved first in human on 2002 by our team. So, our important papers were cited.
Best regards

Round 2
Reviewer 1 Report
.
Author Response
PDF was attached. Thank you so much.

Reviewer 2 Report
The article still has some flaws, to mention: Row 117-118: The tooth "...was stabilized with lower left molar (#46) by enamel bonding agent (Super-Bond®, SUN MEDICAL Co., Ltd. Tokyo)..." Because they used composite to fixed the tooth? Tooth immobilization is bad practice in transplants because it leads to ankylosis, the main cause of failure. When was the fixture removed? Line 233: The authors refers that: However, the timing of root channel treatments for donor tooth with complete roots has not been discussed and not well-documented... That's not correct. What the literature says is that teeth with closed apexes should be performed ER before transplantation.
Author Response
We attached PDF. Thank you so much.

Round 3
Reviewer 2 Report
The corrections that the authors made are in accordance with what was proposed by the reviewer, so the article can be accepted for publication